# Variable Temperature Behaviour of the Hybrid Double Perovskite MA_2_KBiCl_6_

**DOI:** 10.3390/molecules28010174

**Published:** 2022-12-25

**Authors:** Fengxia Wei, Yue Wu, Shijing Sun, Zeyu Deng, Li Tian Chew, Baisong Cheng, Cheng Cheh Tan, Timothy J. White, Anthony K. Cheetham

**Affiliations:** 1Institute of Materials Research and Engineering, Agency for Science, Technology and Research, 2 Fusionopolis Way, Singapore 138634, Singapore; 2Stanford Synchrotron Radiation Lightsource (SSRL), SLAC National Accelerator Laboratory, Menlo Park, CA 94025, USA; 3Energy and Materials Division, Toyota Research Institute, California, CA 94022, USA; 4Department of Materials Science and Engineering, National University of Singapore, Singapore 117575, Singapore; 5School of Materials Science and Engineering, Nanyang Technological University, 50 Nanyang Avenue, Singapore 639798, Singapore; 6Materials Research Laboratory, University of California, Santa Barbara, CA 93106, USA

**Keywords:** hybrid halide perovskite, phase transition, Pb-free

## Abstract

Perovskite-related materials show very promising properties in many fields. Pb-free perovskites are particularly interesting, because of the toxicity of Pb. In this study, hybrid double perovskite MA_2_KBiCl_6_ (MA = methylammonium cation) was found to have interesting variable temperature behaviours. Both variable temperature single crystal X-ray diffraction, synchrotron powder diffraction, and Raman spectroscopy were conducted to reveal a rhombohedral to cubic phase transition at around 330 K and an order to disorder transition for inorganic cage below 210 K.

## 1. Introduction

In the past decades, solution-processable hybrid lead halide perovskites MAPbX_3_ (MA = methylammonium cation, X = Cl, Br, I) have achieved a remarkable power conversion efficiency and photoluminescence quantum yield, finding their applications in many fields such as photovoltaics, X-ray detectors, LED, etc. [1,2,3] The search for lead-free alternatives has been the subject of much research in the last few years. Since 2016, making halide double perovskite via heterovalent substitution of Pb^2+^ by a monovalent cation (M^+^) and environmental benign Bi^3+^/Sb^3+^/In^3+^ provided a plausible approach, where the 3D perovskite framework is maintained by alternating corner-connected MX_6_ and BiX_6_ (or SbX_6,_ InX_6_) octahedra. In the last century, inorganic double perovskites, i.e., elpasolites, have been systematically synthesized, and mostly with ionic Li, Na, F, and Cl which generally give wide bandgaps [4,5,6]. Within just one year, six new double perovskites that can be potential photovoltaic materials were reported—MA_2_KBiCl_6_ [7], MA_2_TlBiBr_6_ [8], MA_2_AgBiBr_6_ [9], Cs_2_AgBiCl_6_ [10,11], Cs_2_AgBiBr_6_ [12], and Cs_2_AgInCl_6_ [13], with optical bandgaps ranging from 1.9 eV to 3.3 eV. Later we discovered Cs_2_AgSbBr_6_ with a bandgap as low as 1.55 eV [14]. They showed remarkably similar physical properties with their lead analogues. 

In this paper, we investigate the variable temperature behaviours of MA_2_KBiCl_6_, the first reported hybrid double perovskite, finding a reversible phase transition from $R\bar{3}m$ to $Fm\bar{3}m$ at 330 K. More complicated low temperature behaviour was detected from variable temperature (VT) single crystal and synchrotron powder X-ray diffraction and Raman spectroscopy. 

## 2. Results and Discussions

### 2.1. High Temperature Phase Transition

At room temperature (RT), the MA_2_KBiCl_6_ crystal structure possesses rhombohedral symmetry$\;R\bar{3}m$ (No. 166), a = 7.8379(2) Å, c = 20.9801(6) Å (CCDC 145389), where the MA cations align along the *c* axis and corner-connecting octahedra tilted with K-Cl-Bi angle 173.04° (Figure 1). Both KCl_6_ and BiCl_6_ octahedra are slightly distorted [7].

Upon heating above 330K, a phase transition to typical halide double perovskite symmetry—cubic $Fm\bar{3}m$ (a = 11.4326(2) Å at 380 K, CCDC 2224436)—is seen and MA cations become disordered, similarly to their MA_2_AgBiBr_6_ counterpart [9] (Figure 1 and Appendix A). The octahedra experience a transition from tilted to regular, yielding contractions in the K-Cl (~2.29%) and Bi-Cl (~1.15%) interatomic distances through the phase transition from 320 K to 340 K. KCl_6_ shows a larger contraction due to the lower coulombic affinity between K^+^ and Cl^−^ compared to Bi^3+^ and Cl^−^, weaker Cl^−^ to Cl^−^ repulsion in the KCl_6_ octahedron due to larger K^+^ size, and less directional ionic bonds making the KCl_6_ octahedron easier to rotate/distort. A similar phenomenon is observed in zeolites when substituting Zn^2+^ with Li^+^ and B^3+^—upon applying pressure, a larger distortion is observed around Li, which has lower valence and larger ionic size [15]. After the phase transition, further increasing temperature did not cause significant changes in bond lengths, lattice parameters, or unit cell volumes.

### 2.2. Low Temperature Behaviour

VT PXRD patterns from 300 K to 12 K did not show any peak splitting or abnormal peak broadening; anisotropic thermal expansion was detected by anisotropic peak shifting (Figure 1c). More detailed analysis was conducted by VT SCXRD. Upon cooling, negative and nonlinear thermal expansion along the c axis and continuous volume reduction can be observed throughout the temperature range, and approximately linear thermal expansion 9.07 × 10^−5^ K^−1^ for the a axis is obtained until 190 K (Figure 2). A sudden change regarding the octahedral tilting and distance of hydrogen bond (Appendix A) below 230 K suggests possible phase transitions [16]. The LT structures are analysed based on the organic molecules and inorganic framework, respectively. 

The octahedral tilting, accompanied with distortion, has increased with lowering temperature, where larger distortion occurs again to KCl_6_ (Figure 2 and Appendix A), for similar reasons to the HT phase: that the larger octahedron with a cation of lower valence is more easily distorted. However, below 230 K, the anisotropy of Cl ellipsoids increases abruptly, where the ratio of the maximum, medium, and minimum values can reach ~ 35:2:1 at 120 K (Figure 2d). Symmetry reduction to other rhombohedral, orthorhombic, monoclinic, and even triclinic space groups have been attempted, but they either gave high residual electron density or failed. Hence, we propose a disordered model under the same space group $R\bar{3}m$ with Cl split into two symmetrical equivalent positions, with each having 50% occupancy (Appendix A and Figure 2e). From SCXRD, we are able to narrow down the transition range to between 210 K to 190 K. 

VT Raman spectroscopy further proves the retention of symmetry, as no extra peaks or peak splitting was observed (Figure 3) [17]. Raman spectra were collected from RT to 120 K at a laser wavelength of 532.05 nm. Four peaks are present from 50 cm^−1^ to 350 cm^−1^, similar to its inorganic analogue, Cs_2_NaBiCl_6_ [17]. The possible assignments of frequencies are (1) the stretching of Cl atoms towards and away from the central Bi/K atoms, (2) 3 Cl atoms away from Bi/K and 3 Cl towards central atoms in the octahedron at the same time, (3) octahedral bending and (4) lattice mode of MA cations, from high to low wave numbers, respectively [18].

Although the MA cations seem to be crystallographically ordered at RT, the short C-N bonds (1.32 Å) suggest otherwise, as the anisotropic thermal ellipsoids for both C and N indicate strong transverse vibrations resulting from relatively high molecular mobility. Upon cooling, the MA molecules tend to be frozen, shown as the increasing crystallographic C-N distances (Figure 1 and Figure 4), which provide the major contribution to the negative thermal expansion of c axis. When further inspecting the ellipsoids, the N vibrations tend to become isotropic, and anisotropy of C also decreases yet is still present, which we speculate is due to the change of MA vibrational modes. Figure 4 provides an illustration, of the possible vibrational modes of the MA molecule. At RT, the centre of mass serves as the centre of liberation, yielding similar ellipsoids for C and N. As temperature is reduced, the liberation centre moves towards N, most likely due to the locking-in of hydrogen bonds between N-H···Cl. Similar behaviours can be expected in the rare earth double perovskites MA_2_KYCl_6_ and MA_2_KGdCl_6_ [19].

## 3. Materials and Methods

Single crystal growth: A saturated solution of MACl (MA: methylammonium), KCl and bismuth acetate (molar ratio 2:1:1) in aqueous HCl (37 wt%) was prepared at 50 °C, then stored at 4 °C. Crystals of millimetre size were obtained using vacuum filtration after 7 days.

Single crystal structure determination from 120 K to 380 K was using an Oxford Gemini E Ultra diffractometer, Mo Kα radiation (λ = 0.71073Å), equipped with an Eos CCD detector. Diffractions at variable temperatures were performed by collecting data from 300 K to 120 K using Cryostream system with N_2_ flow with 40 K steps, then heated up to 380 K for high temperature structure investigation. The crystal stayed under nitrogen flow for a further 30 min at each temperature to allow sufficient equilibration. Data collection and reduction were using CrysAliPro (Agilent Technologies). An empirical absorption correction was applied, and the structure was solved using ShelXS and refined by ShelXL with the Olex2 platform, except for the structure at 180 K, which was solved using Superflip and refined by JANA.

Raman spectra were recorded using a LabRam 300 Raman spectrometer coupled with an Olympus BXFM ILHS confocal microscope with 10 times and 50 times magnification available. The laser wavelength used was 532.05 nm; the laser power was kept at 100mW for the duration of experimentation. The system was calibrated against the 520.5 cm Raman band of a crystalline silicon wafer. The sample holder and cooling stage was a Linkam Scientific DSC600 with associated liquid nitrogen pump. Variable temperature Raman was performed under an air atmosphere. A polycrystalline powder sample was placed in the sample holder/cooling stage which was subsequently sealed. The stage temperature was decreased by 10 K/min, and the sample was allowed to equilibrate at a given temperature for approximately 10 min before the collection of Raman spectra. After data collection at ~120 K, the sample was reheated at 10 K/min to room temperature 296 K, at which point Raman spectra were again recorded.

## 4. Conclusions

The Pb-free hybrid double perovskite MA_2_KBiCl_6_ was found to have interesting variable temperature behaviour. Upon heating to 330 K, a phase transition from $R\bar{3}m$ to $Fm\bar{3}m$ occurs, where the MA cations disorder to the cubic symmetry. At low temperatures, although synchrotron powder diffraction did not indicate any sign of phase transitions, the inorganic cage tended to be disordered below 210 K. The short C-N bond at room temperature is a manifestation of vibrational disorder. 

## Figures and Tables

**Figure 1 molecules-28-00174-f001:**
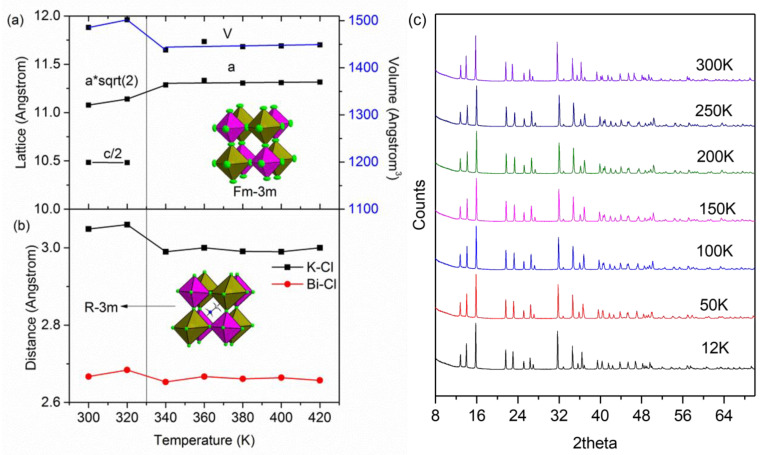
(**a**) Lattice parameters variations as a function of temperature. The lattice parameters of the rhombohedral cell were converted into its equivalent cubic setting. (**b**) Bond distance variations. Octahedral arrangements at both rhombohedral and cubic symmetry are illustrated (MA cation is omitted in cubic cell for illustration purpose. Green: Cl, purple BiCl_6_, brown: KCl_6_. (**c**) The VT synchrotron PXRD patterns.

**Figure 2 molecules-28-00174-f002:**
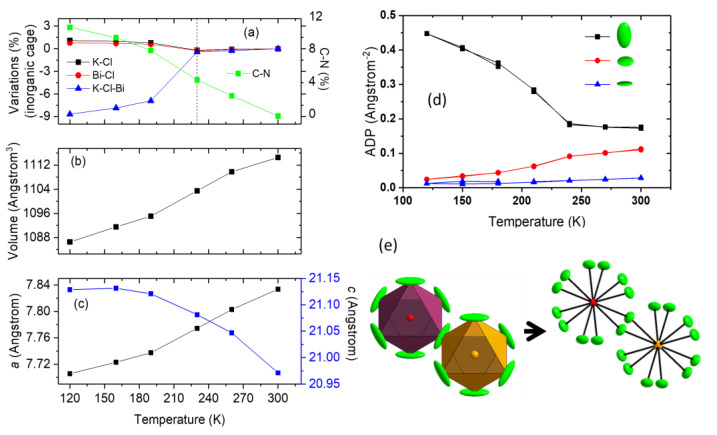
(**a**) Bond length and angle variations as function of temperature during cooling. Left axis: K-Cl, Bi-Cl bond lengths and K-Cl-Bi angle. Right axis: C-N distance. Both y axes show percentage changes from 225 K for easy comparison. (**b**) Volume and (**c**) lattice parameters expansion in response to temperature. (**d**) Cl anisotropic displacement parameters along the maximum, medium and minimum elongation directions. (**e**) The Cl splitting model. Data are from SCXRD.

**Figure 3 molecules-28-00174-f003:**
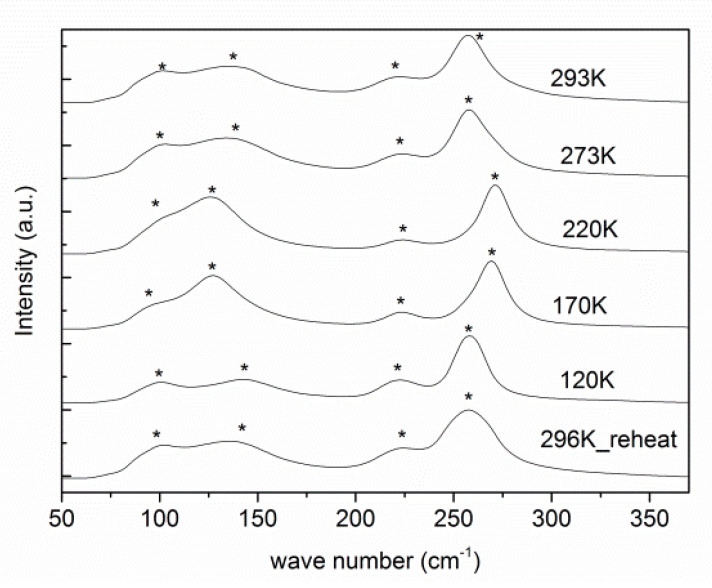
VT Raman spectra during cooling for (MA)_2_KBiCl_6_. The peaks are highlighted by *.

**Figure 4 molecules-28-00174-f004:**
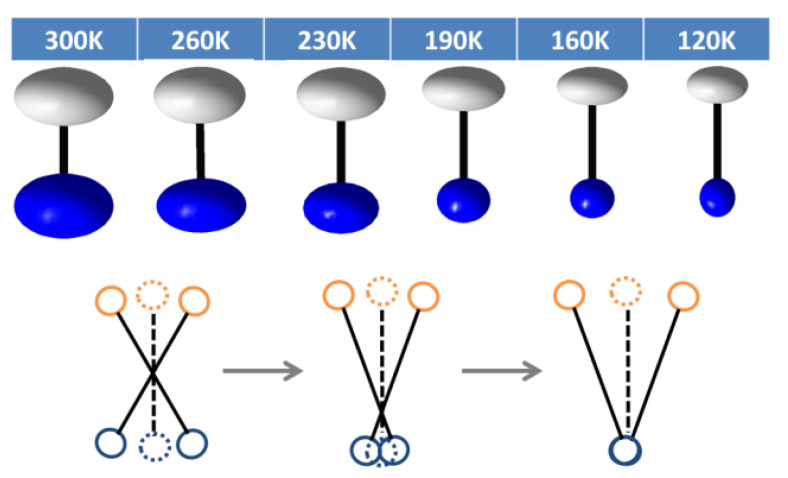
C-N thermal ellipsoids (at 50% probability) and the possible vibration modes of MA at different temperatures. Grey and brown: C, blue: N.

## Data Availability

All data are available upon reasonable request from corresponding author.

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
