# Peer review of "Variable Temperature Behaviour of the Hybrid Double Perovskite MA2KBiCl6"

_molecules, 2022, doi:10.3390/molecules28010174_

Round 1

Reviewer 1 Report

Perovskite-related materials show very promising properties in many fields. Pb-free perovskites are particularly interesting, because of the toxicity of Pb. In this study, hybrid double perovskite MA2KBiCl6 (MA = methylammonium cation) was found to have interesting variable temperature behaviors. Both variable temperature single crystal X-ray diffraction, synchrotron powder diffraction and Raman spectroscopy were conducted to reveal a rhombohedral to cubic phase transition at around 330 K and an order to disorder transition for inorganic cage below 210 K.I recommend this paper to be published Molecules after considering the following points:

1.In page 3, line 80,  A sudden change regarding the octahedral tilting below 230 K suggests possible phase transitions.please provide some references.

2. In page 4, line 106, “VT Raman spectroscopy further proves the retention of symmetry, as no extra peaks or peak splitting was observed (Figure 3).”Please provide references on the “VT Raman spectroscopy ”method.

3. In page 4, line 123,  Figure 4 provides an illustration, of the possible vibrational modes of the MA molecule. Does the vibration of MA have any detailed impact on material stability in practice use?

4. Whats the maximum absorption wavelengths for the cubic phase at around 330 K and the rhombohedral phase below 210 K, separately.

Author Response

We thank the reviewer for the comments.  All the changes based on the reviewer's suggestions have been included and highlighted in the revised manuscript.  Please see the point to point response below:

1: Reference [16] has been included.

2: Reference [17] has been added. 

3:  The vibrational mode of MA cations play a very important role in materials stability. It is strongly coupled with the octahedral tilting in the inorganic cage. Higher vibrational entropy at higher temperature can help to stabilize the structure.

4. We thank the reviewer for the nice suggestions. Knowing the absorption wavelength for different symmetries at different temperatures can certainly be very significant. But due to the instrumental limitations, we are not able to conduct such experiments in our lab. We would search for collaborations in the future to do detailed studies on this topic. 

Reviewer 2 Report

The authors study the variation in the structure of MA2KBiCl6 double perovskite as a function of temperature. They report a phase transition from rhombohedral to a cubic phase at a temperature near 330 K. This confirms the expectation that at higher temperatures the crystal adopts a higher symmetry configuration. The low-temperature behavior is more complicated, and the authors provide some speculation, backed by Raman data, that at lower temperatures, there is a disorder in the position of the Cl atom.

For applications to solar cells, it would be useful to know how the band gap and optical absorption change at the phase transition, particularly the one near 330 K. The authors may want to consider further experimental work to study the variation of the optical properties with temperature.

Author Response

We thank the reviewer for the comments.  Due to the instrumental limitations, currently we cannot do variable temperature absorption spectrum, but we are seeking for collaborations and would conduct such studies in the near future. 

Reviewer 3 Report

The authors have identified a phase change in a lead-free double perovskite, and have analysed the structure of MA2KBiCl6 using a variety of different methods at a range of temperatures. The analysis is thorough and clearly supports the conclusions made. This work would be of interest to the wider research community to aid the study of these new materials. Figures are all clear and appropriate and easy to obtain the data/results from.

Some additional detail could be added to the introduction to discuss previous temperature dependent crystallography studies to provide additional context. Figure S2 has not been referenced within the text.

Author Response

We thank the reviewer for the comments. Figure S2 is now referred in the main text at Page 3, line 81.  

Reviewer 4 Report

The author presented the temperature-dependent structural characteristics in halide double perovskite. K+ and Bi3+ were used to substitute the lead giving rise to double perovskite with wider bandgaps. In the study of the double perovskite MA2KBiCl6, the authors presented two kinds of structural evolution concerning changed temperature as detected by variable temperature single crystal and synchrotron powder X-ray diffraction and Raman spectroscopy. Specifically, the MA2KBiCl6 experienced a phase transition from ?? to ??? at 330K due to the transition of the KCl6 and BiCl6 octahedra. On the other hand, the octahedra tilting below 230K is related to the increase of the anisotropy of Cl ellipsoids. A splitting in the Cl ellipsoid into two symmetrical equivalent positions was proposed to explain the disordered structure. Overall, the investigation of the temperature-dependent behavior of the hybrid double perovskite MA2KBiCl6 and the related analysis of the structural changes are of scientific significance. I have some minor questions regarding the details of the study.

1.       For the high-temperature phase transition, does MA2KBiCl6 recover its rhombohedral symmetry ?? when cooled back to room temperature? Is there any kind of hysteresis if the such measurement has been done?

2.       In the Raman spectra in Figure 3, what kind of lattice vibrations do the four modes correspond to? There seem to be sudden frequency shifts for the Raman modes measured at 220K and 170K. Are these frequency shifts potentially related to the splitting of the Cl ellipsoid? It would be helpful if the authors could elaborate more on this point.

3.       The caption for figure 2c is missing.

Author Response

We thank the reviewer for the comments.  We have revised the manuscript and highlighted in the main text. Please see the point by point response below:

  1. The phase transition is reversible, when cooling to room temperature, it would recover the rhombohedral symmetry.  Unfortunately, we don't have a temperature control unit that is accurate enough to see if there exists a hysteresis. 
  2. We agree with the reviewer that the frequency shift from 220K to 170K is probably related to the splitting of Cl ellipsoids. We have attempted to identify the four peaks, it turned out to be very challenging and each peak may correspond to multiple vibration modes, the details are in the main text, page 4, line 109. 
  3. We are sorry for the mistakes. Caption of Figure 3C is now added at Page 3, line 87.